# Diagnosis of Neonatal Late-Onset Infection in Very Preterm Infant: Inter-Observer Agreement and International Classifications

**DOI:** 10.3390/ijerph18030882

**Published:** 2021-01-20

**Authors:** Gaelle Bury, Stéphanie Leroux, Cristhyne Leon Borrego, Christèle Gras Leguen, Delphine Mitanchez, Geraldine Gascoin, Aurore Thollot, Jean Michel Roué, Guy Carrault, Patrick Pladys, Alain Beuchée

**Affiliations:** 1Department of Neonatology, CHU Rennes, Rennes University, Inserm-CIC 1414, F-35033 Rennes, France; gaelle.bury@chu-rennes.fr (G.B.); stephanie.leroux@chu-rennes.fr (S.L.); alain.beuchee@chu-rennes.fr (A.B.); 2LTSI-UMR_S Inserm 1099, University of Rennes, F-35000 Rennes, France; cristhyne.leon@univ-rennes1.fr (C.L.B.); guy.carrault@univ-rennes1.fr (G.C.); 3Department of Neonatology, University Hospital of Nantes, F-44000 Nantes, France; christele.grasleguen@chu-nantes.fr; 4Department of Neonatology, University Hospital of Tours, F-37044 Tours, France; delphine.mitanchez@univ-tours.fr; 5Department of Neonatology, University Hospital of Angers, F-49100 Angers, France; geraldine.gascoin@orange.fr; 6Department of Neonatology, University Hospital of Poitiers, F-86021 Poitiers, France; Aurore.THOLLOT@chu-poitiers.fr; 7Department of Neonatology, University Hospital of Brest, F-29200 Brest, France; jean-michel.roue@chu-brest.fr

**Keywords:** neonate, sepsis, infection, heart rate variability, bronchopulmonary dysplasia

## Abstract

Background: The definition of late-onset bacterial sepsis (LOS) in very preterm infants is not unified. The objective was to assess the concordance of LOS diagnosis between experts in neonatal infection and international classifications and to evaluate the potential impact on heart rate variability and rate of “bronchopulmonary dysplasia or death”. Methods: A retrospective (2017–2020) multicenter study including hospitalized infants born before 31 weeks of gestation with intention to treat at least 5-days with antibiotics was performed. LOS was classified as “certain or probable” or “doubtful” independently by five experts and according to four international classifications with concordance assessed by Fleiss’s kappa test. Results: LOS was suspected at seven days (IQR: 5–11) of life in 48 infants. Following expert classification, 36 of them (75%) were considered as “certain or probable” (kappa = 0.41). Following international classification, this number varied from 13 to 46 (kappa = −0.08). Using the expert classification, “bronchopulmonary dysplasia or death” occurred less frequently in the doubtful group (25% vs. 78%, *p* < 0.001). Differences existed in HRV changes between the two groups. Conclusion: The definition of LOS is not consensual with a low international and moderate inter-observer agreement. This affects the evaluation of associated organ dysfunction and prognosis.

## 1. Introduction

Late-onset sepsis (LOS), occurring after 48 h, 72 h or 7 days of life according to definitions, is a common complication of prematurity. Despite an improvement in its incidence, 10–25% of very premature infants admitted to neonatal intensive care units will develop LOS, which is responsible for 10–20% of deaths [1]. In addition, LOS has been associated with prolonged ventilatory support and intravascular access, excess bronchopulmonary dysplasia (BPD), necrotizing enterocolitis (NEC), and neurological damage, including intraventricular hemorrhage with impacts on growth and neurodevelopment [2,3,4,5,6,7,8,9]. The use of antibiotics is also potentially harmful. The adverse effects of prolonged antibiotic exposure in preterm infants include antibiotic resistance, fungal infections, NEC, death and possible long-term adverse effects [8,10,11,12,13]. The management of LOS, therefore, requires accurate and early diagnosis, but its definition is not consensual [14].

In adults, sepsis is no longer defined in reference to the systemic inflammatory response but as the dysfunction of at least one vital organ caused by a deregulated host response to infection [15]. The 2005 international definition of pediatric sepsis based on an adaptation of the SIRS and the assessment of organ dysfunction excluded preterm infants [16]. In the neonatal period and particularly in preterm infants, vital and immunological functions are immature, and it is, therefore, difficult to define a deregulated host response to infection. Signs of infection, whether clinical or biological, are often discrete, not very sensitive and not very specific with the possibility of rapid clinical deterioration. The concept of “organ dysfunction” is not defined in this population with no available consensus reference or established score to describe the physiological changes associated with LOS during the neonatal period, particularly in preterm infants [14,17]. The microbiological diagnosis is also often unreliable in preterm infants. The time to obtain the result is often long, and it is often not easy to correlate the clinical picture with the identified bacteria. Since 2008, the bacteriological diagnostic criteria proposed in neonatology by the American NHSN network has required either at least one positive blood culture with a pathogenic germ, or two positive blood cultures, taken on two separate occasions, in the case of a microorganism considered to be a skin saprophyte. The bacteria most associated with LOS are the coagulase-negative *Staphylococci* (CoNS), but blood volume on one hand, and the need to start antibiotic therapy without delay on the other, only rarely allow two blood cultures to be taken before treatment is initiated [3]. C-reactive protein (CRP), procalcitonin (PCT), leukocytosis and thrombocytopenia are commonly described as sepsis biomarkers but are also often late, nonspecific and non-sensitive [18,19]. In summary, all the elements of the diagnosis, whether clinical, biological or microbiological, remain debated [14,20]. It is now recognized that there is no consensus definition of LOS and that this is required to improve diagnosis and treatment and limit the associated sequelae. The choice of an operational definition is, however, difficult because it necessarily depends on the expected precocity, sensitivity and specificity.

The study of heart rate variability (HRV) may be useful to identify organ dysfunction associated with LOS [21,22,23,24,25]. HRV has been used to detect LOS in a randomized clinical trial involving 3003 preterm infants, which has shown a reduction of associated LOS mortality by 40% in very low birth weight preterm infants [26].

In the present study, we wanted to assess the concordance of LOS diagnosis between experts in neonatal infection and international classifications and to evaluate the potential impact of diagnosis certainty on heart rate variability and rate of “bronchopulmonary dysplasia or death” in a prospective cohort of preterm infants treated with at least five days of antibiotic therapy for LOS.

## 2. Materials and Methods

The data used in this study was part of the database of the Digi-NewB (https://www.digi-newb.eu/) observational cohort (NCT02863978, Horizon 2020 European project GA-689260). This cohort prospectively included newborn infants hospitalized in the neonatal units of six university hospitals in the western region of France (University Hospitals of Rennes, Angers, Nantes, Brest, Poitiers, and Tours) in 2017–2019. The collection of the data was carried out after approval by the ethics committee (CPP Ouest IV 34/16), the national agency for the safety of medicines and health products (Authorization number: 2016062400181) and informed parental consent. The Digi-NewB project is aimed at improving clinical intervention in neonatal units by developing a new generation of non-invasive perinatal health monitoring based on a multi-parametric approach. The objective was to assist clinicians in their decision-making with regard to the risk of LOS.

### 2.1. Study Population

This study included very preterm infants (VPI) born before 31 weeks gestational age (GA) between 1 November 2016 and 3 October 2019. All the newborns included in this study were clinically suspected of LOS and received more than five days of antibiotics (or die before the course is completed), beginning at least 72 h after birth. For each VPI, only the first episode of LOS was studied.

Maternal, obstetrical, and birth data, as well as the clinical, biological, and microbiological data surrounding LOS suspicion, were prospectively collected. The presentation of all these data was prepared in a user-friendly presentation by the investigators to facilitate the case analyses by the experts. All the characteristics of each very preterm infant were checked, and curves presenting the evolution of the studied parameters were prepared. Then, the two main investigators classified each case according to international classifications from the National Institute for Child Health and Human Development (NICHD) [8], the Canadian Neonatal Network (CNN) [27], the US Centers for Disease Control and Prevention (CDC) [28], and the German National Reference Centre for Nosocomial Infection Surveillance in Neonatal Intensive Care Units (NEO-KISS) [29,30]. The LOS were further characterized by the investigators as central line-associated bacterial infections documented on blood cultures or not (CLABSI or CLASS) and infections documented on blood cultures or not without the involvement of a central catheter for ether laboratory confirmed bloodstream infection (LCBI) or clinical sepsis (CSEP). Bacterial infection was considered documented on blood culture when one or more pathogenic or commensal bacteria was identified on blood cultures without being related to an infection at another site. The central line was considered associated with the infection if in place for at least 48 h or removed for less than 24 h at the time of LOS suspicion. The CSEP or CLASS classifications were used in cases of clinical symptoms suggestive of LOS with no microorganisms detected in the blood or at any other site. The two principal investigators independently characterized each inclusion and then confronted each other in case of disagreement in order to reach a consensus before submission to the experts of the adjudication committee.

### 2.2. Variables Studied

Maternal, obstetrical and birth data, as well as the data collected around LOS suspicion, were provided to the adjudication committee. These data had been prospectively collected continuously in 6 h periods. The period surrounding the infection was defined as five days before and five days after the start of antibiotic administration (T0). The 13 clinical parameters used were collected from electronic case report forms or from analyses of heart rate, respiratory rate and transcutaneous oxygen saturation traces obtained from the monitors. All occurrences of abnormal clinical or biological signs were annotated with regard to T0.

The clinical signs studied were as follows: cardio-respiratory events, bradycardias, increase in oxygen or ventilatory assistance, peripheral hemodynamic (cutaneous refilling time >3 s and/or occurrence of pallor or marbling), central hemodynamic (heart rate, blood pressure, need for hemodynamic drug or volume expansion), digestive symptoms (abdominal distension, feeding cessation), body temperature (<36 °C or >38 °C or unstable), abnormal neurological examination, and presence of painful symptoms. Reported cardiorespiratory events were related to the number of apneas and/or bradycardia and/or desaturation. Bradycardia was defined using two levels of severity as a heart rate <100 beats/min for >5 s or <80 beats/min for >10 s (expressed as a number/h for each level of severity). The number of cardio-respiratory events was expressed as an absolute value per 6 h period.

The increase in oxygen or respiratory assistance was analyzed through the presence of signs of respiratory struggles, the inspiratory oxygen fraction (Fi02, figure expressed in %), the positive expiratory pressure (PEP, figure expressed in cm of water) or the mean pressure in case of assisted mechanical ventilation. In order to have a more global appreciation of the evolution of the respiratory support in the study and not to rely on a single FiO2 or respiratory pressure figure, we have proposed a respiratory score, the “FiO2-P score”. This score does not depend on the type of respiratory assistance (e.g., continuous positive airway pressure, intra-tracheal mechanical ventilation). The first step is to score the FiO2 in steps corresponding to a number of points: from 21 to 25% inclusive = 1 point, 25–30% = 2, 30–35% = 3, (…), 95–100% = 16 points. In a second step, we quoted the PEP (or mean respiratory pressure value in case of mechanical ventilation) values by steps: PEP (or mean pressure in case of mechanical ventilation) at +4 cm H20 = 1 point, +5 = 2 points, (…), +10 = 7 points. Finally, the FiO2-P score is the sum of the FiO2 score and the PEP score.

Digestive signs collected included feeding stops of more than 6 h and abdominal distension described as at least “moderate with visible intestinal loops”. The date and duration of the feeding stop were specified. Episodes diagnosed as necrotizing enterocolitis (NEC) following Bell classification at the time of LOS suspicion were also considered and reported in the table. Abnormal neurological signs were defined as hypoactivity or altered state of consciousness. Painful symptoms were rated by the nurses on an EDIN or Comfortneo scale depending on the context [31,32]. For EDIN >3 or a Comfortneo score >12, the research nurse reported the presence of pain symptoms over the corresponding 6 h period. The appearance of hyperglycemia (absolute value and qualitative assessment relative to the threshold of 140 mg/L) and/or glycosuria (retained if greater than ++ on the urine test strip) was noted.

The “number of associated signs” was counted among the following signs: thermal instability, cardiorespiratory events, neurological signs, abnormal peripheral hemodynamic, unusual episode of hyperglycemia, increased respiratory assistance, tachycardia, and tachypnea.

The analyzed inflammatory markers included C-reactive protein (CRP, expressed in mg/L) and procalcitonin (PCT). For all biological variables, we selected the most extreme value for the period of time between T0 − 12 h and T0 + 60 h [18]. The micro-biological parameters studied included blood cultures, cerebrospinal fluid (CSF), tracheal aspirations, and catheter cultures. Bacteriological examination of urine was not included in the study as this was not standard practice in the neonatology units involved in the study. Samples taken at T0 + 48 h were taken into account. Isolated microorganisms were identified as pathogenic or contaminant from the NHSN organism list [29]. The delay between the blood sample taken for microbiology (corresponding to clinical suspicion) and the start of antibiotic therapy was expressed in hh:mm.

Bronchopulmonary dysplasia (BPD) was defined by the persistence of respiratory support (oxygen and/or continuous expiratory pressure and/or high-flow nasal cannula) at 36 weeks GA [33]. The diagnosis of BPD was assessed during the prospective follow-up of the cohort; this information was not made available to the adjudication committee. If a death occurred during hospitalization, it was recorded with its date.

### 2.3. Adjudication Committee

An independent committee was composed of 5 high-level experts well recognized for their in-depth knowledge of LOS, with four experts in neonatology and one expert in microbiology involved in the evaluation of nosocomial infections occurring in neonatology. These experts were asked to independently qualify their assessment of each recorded case according to a three-level Likert scale as “certain or very probable”, “probable,” or “doubtful” infection. The descriptive files provided included the clinical, biological and bacteriological data of all patients grouped in a table together with a slide show containing the detailed individual clinical course of each patient. Temperature, cardiorespiratory events frequency, bradycardia frequency, mean heart rate, mean respiratory rate, FiO2-P respiratory score were presented as individual curves with a time scale based on 6 h-periods (example in Figure 1). A documentary database was also given [14,27,28,29].

The information concerning the prognostic composite criteria was blinded to the expert as well as the results of the HRV analysis.

At the end of the adjudication process, the cases were post hoc classified by the investigators into two groups. Newborns were considered “infected” if there were at least 3 “probable” or “very probable” ratings out of the five and “doubtful” in other cases.

### 2.4. Organ Dysfunction Assessment: Study of HRV

The chain of treatment from the acquisition of the ECG to the computation of the HRV features has been described in detail in C Leon et al. [34]. In summary, the ECG from the infants were obtained with a sampling rate of 500 Hz. The RR intervals (cardiac cycle length) were detected with algorithms dedicated to preterm infants, and a sliding window of 30 min [35], with no overlap, was applied to calculate the RR beat-to-beat time-series and from it, all the HRV parameters were calculated. Heart rate variability (HRV) provides indirect insight into normal and abnormal control and reactivity of heart rate in a systemic approach. HRV was assessed using different methods (i.e., time-domain, frequency-domain, nonlinear methods, graph theory) in order to characterize HRV in terms of amplitude, periodicity, regularity, complexity, and underlying structure of the organization. The time-domain parameters calculated were the mean of the RR intervals (meanRR); the standard deviation (sdRR); the root mean square of successive RR intervals (rMSSD), which characterize short-term HRV; the maximum and minimum value for the RR intervals in the time series (maxRR and minRR, respectively); the characteristics of the distribution of RR duration (skewness, kurtosis); and the acceleration and deceleration capacity of the heart rate (AC, DC). The frequency-domain parameters calculated were the low-frequency power (LF), with limits 0.02–0.2 Hz, the high-frequency power (HF), with limits 0.2–2 Hz. We also calculated these features in normalized units (LFnu = LF/LF + HF and HFnu = HF/LF + HF) and the LF/HF ratio. The nonlinear parameters calculated were the sample entropy (SampEn) and approximate entropy (ApEn), which estimates the level of regularity and predictability of the signal; the coefficients α1 and α2, obtained from the detrended fluctuation analysis of the time-series, and which represent, respectively, the short-range and long-range fractal correlations of the signal; and the parameters SD1 and SD2, derived from the Poincaré plot, and which reflect the short and long term variability, respectively. The graph theory parameters calculated included vertical and horizontal visibility studies [34,36]. The visibility graph (VG) is a network analysis of time-series; it converts the series into a graph that inherits several of its properties, transforming each data point in the time-series into a node that is vertically or horizontally connected with other nodes. The parameters studied were the average degree (MD_VG and MD_HVG, respectively), the clustering coefficient (C_VG and C_HVG, respectively), the transitivity (Tr_VG and Tr_HVG, respectively) and the assortativity (r_VG and r_HVG). These measures quantify the network organization, connectivity, and complexity.

All the measurements made were standardized and expressed as relative values using the distribution observed outside the infectious episode as a reference value for each infant, and each parameter studied. The reference values were calculated over a period of 12 h outside the infectious episode (at least three days before or if unavailable four days after T0) and outside the period of initial adaptation to extra-uterine life (outside the first three days of life). These periods were used to center-norm all the HRV values studied during the infectious period (between T0 − 6 h and T0 + 6 h) [34]. The results are therefore expressed as Z scores (i.e., (observed value—mean of the reference period)/standard deviation from the reference period).

### 2.5. Statistics

The time windows used for analyses were chosen from the literature review [18,28,34]; they are presented in Figure 2.

The results are presented in number and % or median (interquartile range, IQR) or mean (standard deviation, SD) depending on the distribution of the variables. Comparisons were made with Mann–Whitney U-tests and Mac Nemar’s Chi 2 test using Statistica 13.2 software (StatSoft Inc., Tulsa, USA). In order to answer our main hypothesis, we studied inter-rater reproducibility by comparing the LOS evaluations of each expert 2 to 2 by Cohen’s Kappa tests. A Fleiss kappa was also used to calculate the overall reproducibility of the LOS ranking by the 5 experts. This work was also carried out for the international classifications. We also calculated the Cohen Kappa between the synthetic general classification resulting from the analysis of the results of the adjudication committee and the results of the international classifications. The classification of LOS into two groups, “infected” vs. “doubtful”, was compared to the international classifications CDC, CNN, and NEO-KISS (infected or not). The synthetic classification into two LOS groups was not compared to the NICHD classification, which defines three groups (i.e., certain, probable, doubtful). Kappa values were expressed in absolute values and interpreted according to the Landis and Kock rating scale (excellent inter-observer agreement if Kappa greater or equal to 0.81, good between (0.80–0.61), moderate between (0.60–0.41), poor between (0.40–0.21), bad between (0.20–0), very bad (<0). HRV parameters were expressed in Z-score, as explained above. Comparisons were made using Mann and Whitney’s U-test with Bonferroni correction for multiple comparisons at the alpha * = 0.05 threshold (corresponding to a significance of 0.95 and a *p* value of 0.0033 after correction).

## 3. Results

### 3.1. Population

During the study period, 60 preterm infants born before 31 weeks gestation who had received more than five days of antibiotics were included in the Digi-NewB cohort. Twelve patients were excluded from the current study because nine had received antibiotics started before 48 h of life, and three had not enough data available at the time of the suspected infection to be reliably characterized. One infected preterm infant who died 24 h after the start of antibiotics was included in the study. The characteristics of the 48 infants included in the study population are presented in Table 1. The median gestational age at birth was 26 weeks + 3 days (IQR: 25 w + 2d–27 w + 5d) with a birth weight of 785 g (IQR: 726–911) and a sex M/F ratio of 1.4. Adaptation at birth was good, with an Apgar score greater than 7 at 5 min of life for 38 infants (79%). Early-onset sepsis was suspected at birth in 14 infants (29%) for whom antibiotics were stopped before three days of life after confirmation of the absence of infection.

Cardiorespiratory events included apnea and/or bradycardia and/or desaturation. They were noted in 43 infants. They occurred or increased within 24 h before and 12 h after the start of antibiotics in 34 cases (69%). All but one infant had respiratory support prior to suspected LOS, with increased respiratory support around suspected infection for 15 of these infants, of whom intubation for mechanical ventilation was required for seven. The median maximum FiO2 at the time of infection was 30 (IQR 21–48)%. Feeding discontinuation was carried out in 23 children for a median duration of 4 (IQR: 1–13.5) days. The diagnosis of NEC was associated with LOS in 8 patients (17%).

The CRP was available in all but one patient. The median maximum value around clinical suspicion time was measured at 17 (IQR: 5–40) mg/L and remained less than or equal to 10 mg/L for 18 patients (37%). The PCT was available for 27 patients and was less than or equal to 0.5 ng/L in only two cases.

The blood sample prescribed for suspicion of LOS was drawn at an average postnatal age of 7 days + 11 h (IQR: 5 d + 7 h–10 d + 16 h). The central line was considered associated with the infection in 45 cases (94%). Blood cultures were always taken in the 48 h before the start of antibiotic, with 13 infants having had two or more blood cultures. The delay between blood culture and the start of antibiotics was 2 h (IQR: 00:18–04:46). Blood cultures were negative for 15 patients (31%). Among them, bacteria were identified in another site in 4 cases: two isolates of *Staphylococcus aureus* and one *Klebsiella pneumoniae* on tracheal aspirations and one pyelonephritis with *Enterococcus faecalis.* A bacterial pathogen was isolated from blood cultures in three cases (6%): *Enterococcus faecalis*, *Pantoea agglomerans*, *Staphylococcus aureus*. At least one CoNS was isolated for the remaining 30 cases. A single CoNS was found on two successive blood cultures in 9 cases, (*Staphylococcus epidermidis* (*n* = 5), *Staphylococcus haemolyticus* (*n* = 3), *Staphylococcus capitis* (*n* = 1)) and on a single blood culture in 13 cases: (*Staphylococcus epidermidis* (*n* = 5), *Staphylococcus haemolyticus* (*n* = 4), *Staphylococcus capitis* (*n* = 2), *Staphylococcus warneri* (*n* = 1)). At least two different CoNS were identified on blood cultures in 8 cases (*Staphylococcus epidermidis* (*n* = 8), *Staphylococcus haemolyticus* (*n* = 6), *Staphylococcus capitis* (*n* = 2), *Staphylococcus warneri* (*n* = 2), *Staphylococcus hominis* (*n* = 1)). The only one lumbar puncture performed was negative.

### 3.2. Classifications

The number of “validated” LOS cases varied according to definitions: 13 patients were considered LOS according to CDC criteria, 28 for NICHD, 41 for NEO-KISS, and 46 for CNN. For NICHD, the 28 LOS cases were classified as “certain” for 22 (79%) and “probable” for six (21%). The average rate of agreement between the three classifications CDC, CNN and NEO-KISS for the 48 suspected LOS cases was 54% (IC 95%: 23–85%), for a Fleiss kappa coefficient of −0.08 (IC 95%: −0.29–0.13, *p* < 0.0001), i.e., a very poor inter-observer agreement (Table 2). The number of CLABSIs also varied according to the definitions: 12 for CDC, 30 for CNN and 36 for NEO-KISS.

The five experts’ assessment was unanimously consistent for 11 “certain” and four “doubtful” LOS cases. The median rate of agreement among the experts was 62% (95% CI: 55–69%), for a Fleiss kappa coefficient measured at 0.41 (0.34–0.49, *p* < 0.0001), i.e., moderate inter-observer agreement (Table 3). In the post hoc classification into two groups, resulting from the experts’ classification, the infection was considered “certain or probable” in 36 newborns (75%) and “doubtful” in 12 newborns (25%). The characteristics of the two groups are detailed in Table 1. There were no significant differences between the groups in terms of gestational age (26.2, IQR: 25.2–27.4 vs. 26.5, IQR: 25.8–29.14 SA), birth weight: (785, IQR: 712–912 vs. 815, IQR: 735–955 g), respiratory resuscitation measures used at birth or initial surfactant requirements. However, there were tendencies towards a lack of inter-group similarity in terms of sex ratio (sex ratio M/F of 1.6 vs. 1, NS), date of start of antibiotic (8.1, IQR: 5.6–10.4 vs. 6.2, IQR: 4.8–6.2 days, NS), need to use chest compressions at birth (*n* = 0 vs. *n* = 3, *p* < 0.05), and the number of cases who received antibiotics during the first two days (13 (36%) vs. 1 (8%), *p* = 0.07). At the time of the initiation of antibiotics, thermal instability, abnormal peripheral hemodynamic with prolongation of cutaneous refilling time, and change in respiratory frequency were more frequent in the “certain or probable” group than in the “doubtful” group. CRP was also higher in the “certain or probable” group (24 (IQR: 11–46) mg/L vs. 2 (IQR: 0–6) mg/L, *p* < 10^−4^) with values >10 mg/L for 80% of the “certain or probable” group (*n* = 29) vs. 8% in the “doubtful” patients (*n* = 1).

Bronchopulmonary dysplasia (*n* = 22) or death (*n* = 9) occurred in 31 patients (65%). The rate of “bronchopulmonary dysplasia or death” was higher in the “certain or probable” group than in the “doubtful” group (*n* = 28 (78%) vs. 3 (25%), *p* < 0.001, Odds Ratio = 10 with CI 95%: 2–28). No deaths occurred in the “doubtful” group. The rate of “bronchopulmonary dysplasia or death” was also significantly higher with the NEO-KISS classification (*n* = 29 (71%) vs. 2 (29%), *p* < 0.05, Odds Ratio = 6 with CI 95%: 1.02–36), but not with the other classifications studied.

### 3.3. Evaluation of Organ Dysfunction by HRV

The HRV study carried out over the 12 h period surrounding T0 (between T0 − 6 h and T0 + 6 h) was available in 40 infants (30 in the “certain or probable” and 10 in the “doubtful” group. The significant results are presented in Table 4. Infection was, as expected, associated with significant changes in HRV in the “certain or probable group”. This mainly concerned a decrease in short-term respiratory mediated HRV (rMSSD, HF, HFnu and SD1) and complexity of HRV (SampEn, ApEn), which were associated with an alteration of the general structure of HRV (alpha 1 and TrVG) and of the distribution of cardiac cycle lengths (Kurtosis). It is to note that the sole difference noted in the doubtful group between the infected and control periods was for SampEn and that there was a tendency for the median amplitude of changes in HRV to be lower in “doubtful” than in “certain or probable group”. The differences between the two groups were statistically significant for HFnu and Kurtosis.

## 4. Discussion

This retrospective analysis of the prospective multicenter Digi-NewB cohort confirms that the definition of LOS in very preterm infants is not consensual with a low international and moderate inter-observer agreement. This affects the evaluation of associated organ dysfunction and prognosis. Indeed, if we consider HRV as a physiological marker of organ dysfunction associated with LOS in preterm infants, there is a difference in the extent of impaired function between the two groups defined according to the certainty of the diagnosis. Differences also exist in short-term prognosis between the groups following evaluation by the adjudication committee with an important excess of BPD or death when the diagnosis was considered certain or probable. This could suggest that the current criteria used to assess the diagnosis of LOS leads to a diagnosis too late.

The analysis performed by the expert adjudication committee takes into account all clinical, biological and microbiological signs collected prospectively and sampled over time for each case. This was done in parallel with the use of well-recognized international classifications and independently of the investigators. Moreover, the evaluations were blind to HRV analyses or to the occurrence of death or BPD. This study has, however, some limitations. The criterion for inclusion (i.e., intention to treat at least five days of antibiotics) depends on a medical decision that is not evidence-based and, therefore, likely to vary between clinicians. All clinical signs were prospectively collected, but some criteria as temperature instability, abdominal distension or abnormal peripheral hemodynamic were not based on objective measures. Finally, the classification into two groups was only based on the evaluation by the adjudication committee and is therefore not repeatable. This has, however, been useful to demonstrate the impact of diagnostic certainty on HRV and on the rate of BPD or death.

It seems likely that the multiplicity of international definitions contributed at least in part to the differences in assessment by the experts [14,37]. The potential reasons for the observed inconsistency of the different international classifications in our study are numerous. Variations in the international classifications imply many criteria: clinical criteria used in the definition of sepsis; time at which sepsis occur; interpretation of contaminant or polymicrobial cultures; pathogen isolated from another site than blood or CSF; use of the duration of therapy as a criterion for the diagnosis; cutoff values for sepsis biomarkers; and subclassification of sepsis with regard to NEC, pneumonitis or meningitis. For example, the clinical presentation of what has been termed NEC can be highly variable, with signs and symptoms that are often indistinguishable from sepsis. In our study, we did not introduce specific rules to distinguish between LOS and NEC. One of the consequences of this is that the comparison between published studies is often difficult or unreliable. In our study, the best concordance between the adjudication committee and international classifications was obtained with the NEO-KISS classification with a moderate concordance and an agreement rate of 81%. In addition, we observed a similar impact on heart rate variability and the rate of “BPD or death” with the adjudication committee and NEO-KISS classifications. Therefore, it appears that NEO-KISS is likely to be an appropriate classification to use for the neonatal units involved in the study.

The clinical signs are variably integrated into the criteria for LOS diagnosis. The variability of clinical LOS expression is high and clinical signs are generally considered to be nonspecific and discrete, especially in the early stage of the disease [11]. In our study, collecting 13 clinical criteria, 98% of the children presented at least two signs and 55% at least three signs. The number of associated signs, as identified in this study, however, does not seem to have a high diagnostic value because it did not differ between the two groups. The most frequent signs were cardio-respiratory events, thermal instability, increased respiratory rate and respiratory support. This is coherent with the main identified informative clinical variables in the literature, which were abnormal heart or respiratory rate, abnormal temperature, cardio-respiratory events, abnormal peripheral hemodynamic and skin color, neurological and gastrointestinal symptoms, and hypotension [19,20,24,38,39,40,41,42]. In the current study, we found that thermal instability, abnormal peripheral hemodynamic with prolongation of cutaneous refilling time, and change in respiratory frequency were more frequent in the “certain or probable” than in the “doubtful” group. As the “certain or probable” LOS was associated with poor short-term prognosis, this suggests that these signs occur at an already late-stage of the disease. In the current study, 11 newborns presenting clinical signs and culture-negative suspicion of LOS were included. CNN and NEO-KISS, but not the CDC or NICHD, consider “clinical sepsis” for newborns with clinical signs who have received more than five days of antibiotics with no microorganisms identified. NEO-KISS points out that this situation is close to cases where the blood culture was positive for CoNS. These considerations account for much of the variation in the number of validated LOS between international classifications.

In the neonatal period and with the currently used methods, the early diagnosis of LOS cannot only rely on microbiology testing. In our study, we only identified six infants with bacterial pathogens (three positive blood cultures). All the other isolated bacteria were CoNS. There was only one case of urosepsis, but as suprapubic aspiration is invasive, it was not routinely performed in the neonatal units involved in the study. A risk of urosepsis of at least 10% has been reported in the neonatal period in at least one study [43].

As in all the studies performed in NICU, nearly all the newborns included had a central venous catheter-associated with the LOS suspicion, which is known to be an independent risk factor for CoNS septicemia [20,44]. CoNS was identified in blood cultures for 62% of the preterm infants included in our study. This frequency is close to that reported in studies carried out in industrialized countries, where CoNS infections account for between 53% and 78% of LOS [11,18]. In our cohort, nine patients presented the same CoNS in several samples, which, in the presence of clinical signs, allowed the diagnosis of LOS according to all the international classifications. As CoNS is both the most common contaminant of neonatal blood cultures and the most common pathogen identified in LOS, additional clinical or biological criteria are usually required to confirm the diagnosis. These additional criteria vary between the classifications, and this is an important factor that explains the low concordance between the international classifications.

Certain biological markers can assist in the clinical diagnosis of LOS but are often within the normal range at the time of suspicion. They are not included in the CDC and CNN definitions because of their low diagnosis value at the time of suspicion. For example, CRP levels may not increase or may increase only slowly, in some infected infants, particularly in very preterm infants with CoNS LOS [45]. In our study, the maximum value of CRP surrounding the time of suspicion was less than or equal to 10 mg/L for 18 patients (37%). From a review of 28 articles involving 2661 children, Ruan et al. described the simultaneous measurement of CRP and PCT in neonatal sepsis, with a sensitivity of 0.71 (0.63, 0.78) and 0.85 (0.79, 0.89), respectively, and an AUC of 0.85 (0.82, 0.88) for CRP and 0.91 (0.89, 0.94) for PCT [46]. For this meta-analysis, the best thresholds are 0.5–2 ng/mL for PCT and a threshold value >10 mg/L for CRP, but for other authors, it would seem more relevant to follow the kinetics in relation to the baseline value rather than a threshold value that is not consensual [45]. Biomarkers used in neonatology have traditionally focused on detecting early inflammatory response to sepsis rather than organ dysfunction. This is in contrast with the proposal of the sepsis 3 group in adults and with the proposal of the “nSOFA” score specific to neonatology, which focuses on organ dysfunction to diagnose sepsis [47].

In our study, LOS is, as expected, associated with significant changes in HRV. This reflects an alteration in heart rate regulation [23]. There was also a significant difference in HRV assessment of the organ dysfunction between “certain or probable” and “doubtful” LOS groups reflecting a gradient of change in the HRV according to the degree of diagnosis certainty. A significant difference also existed in both groups with the control period. This may be due to the inclusion in the doubtful group of patients in an early phase of the infection. In this hypothesis and from our results, a loss of HRV complexity would be observed in the early stage of sepsis (decrease in SampEn). This would be followed by the appearance of a decrease in respiratory-mediated HRV (rMSSD, HF, HFnu and SD1) associated with an alteration in the general structure of the HRV (alpha 1, TrVG and kurtosis). Since it is known that the characteristics of HRV are not always specific to the infection [48], one would expect low specificity in the early phase, which increases during the course of the infection.

The incidence of “BPD or death” was clearly more frequent in the “certain or probable” group than in the “doubtful” group (78% vs. 25%, *p* < 0.001). Shah et al. demonstrated the impact of LOS on the occurrence of death or BPD in a retrospective analysis of a cohort, including 7500 babies born before 32 weeks GA. In this study, the odds ratio (OR) of “BPD or death” was higher in children with Gram-negative LOS (OR 2.79, 95% CI 1.96–3.97) and Gram-positive LOS (OR 1.44, 95% CI 1.21–1.71) compared to children without LOS [31]. Jung et al. also found an increased BPD risk with the number of LOS episodes presented during hospitalization. They emphasized that this association does not necessarily mean that the infection is directly responsible for the risk of BPD. Indeed, these babies were more often treated with prolonged mechanical ventilation with an inevitable increase in ventilator-induced lung injury [49].

## 5. Conclusions

This study points out important limits and heterogeneities in the current definition of LOS and confirms the low diagnostic values of clinical, biological, and microbiological criteria for early diagnosis of LOS. It is, however, necessary to get screening criteria to improve care in NICU. Based on the results of the present study, we believe that a multiparametric approach integrating clinical risk stratification and heart rate variability could be a good way to characterize the different stages of organ dysfunction associated with LOS. This could be done using a two-step approach with a sensitive index to suspect infection early and a specific index to confirm the diagnosis later.

## Figures and Tables

**Figure 1 ijerph-18-00882-f001:**
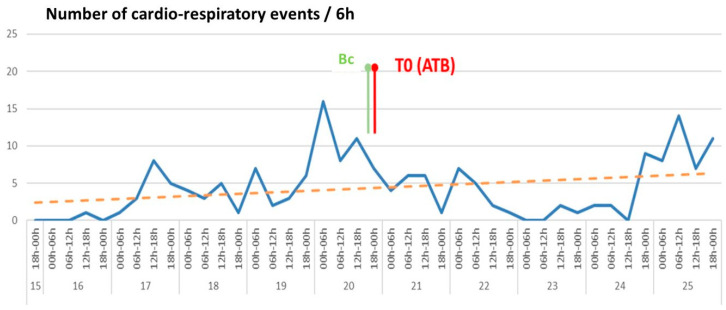
Number of cardio-respiratory events presented by 6 h period (blue line) for one very preterm infant. This figure gives an example of the user-friendly presentation of the results to the adjudication committee. Bc: time of first blood culture, which corresponds to the time of clinical suspicion; T0 (ATB): time of first antibiotic administration; dotted orange line: linear regression of the tendency during the study period. Vertical axis: number of cardio-respiratory events/6 h. Horizontal axis: date and 6 h-time intervals.

**Figure 2 ijerph-18-00882-f002:**
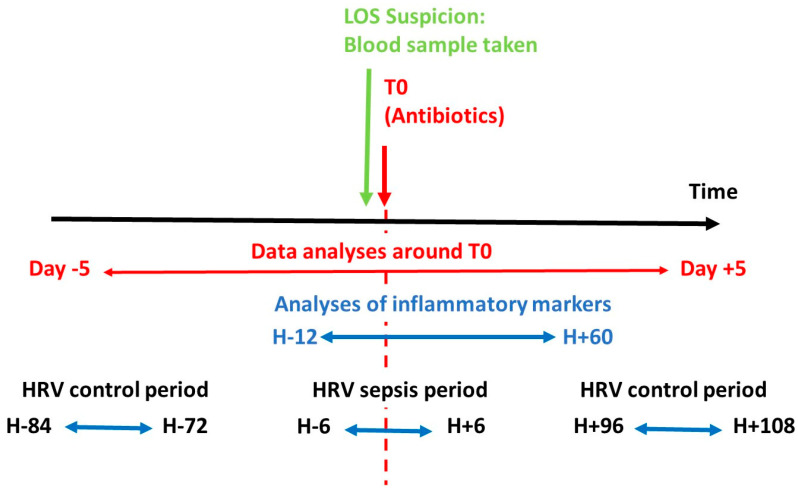
Time windows used for analyses. T0: start of administration antibiotics; period around infection was defined as five days before and five days after T0; the most extreme value of inflammatory markers were obtained between T0 − 12 h and T0 + 60 h; HRV: heart rate variability was studied on 12 h-period between T0 − 6 h and T0 + 6 h with reference values calculated over a period of 12 h outside of infectious episode (at least three days before or four days after T0).

**Table 1 ijerph-18-00882-t001:** Clinical characteristics.

	Certain or Probable (*n* = 36)	Doubtful (*n* = 12)	Total (*n* = 48)
Characteristics of Newborns: n (%) or Median (IQR)			
Birth weight (g)	785 (712–912)	815 (735–955)	785 (726; 911)
Birth term (weeks)	26.5 (25.3–27.4)	26.5 (25.8–29.1)	26.4 (25.3;27.7)
Sex male	22 (61)	6 (50)	28 (58)
Apgar >7 at 5 min	29 (80)	9 (75)	38 (79)
Care in the Birth Room: n (%)			
No special measures	1 (3)	2 (8)	3 (6)
Ventilation only	12 (36)	3 (25)	15 (31)
Ventilation and surfactant	22 (61)	7 (58)	29 (60)
Surfactant without ventilation	1 (2)	0 (0)	1 (2)
Chest compression	0 (0)	3 (25)	3 (6)
Maternal or Obstetrical Data: n (%) or Median (IQR)			
Maternal age (years)	30 (26.75–35.25)	31.5 (28,5–34)	31 (27–34)
Multiplicity	5 (14)	3 (25)	8 (17)
Antenatal corticosteroid therapy	33 (92)	12 (100)	45 (94)
Cesarean section	20 (56)	9 (75)	29 (60)
Prolonged membrane rupture >12 h	9 (25)	3 (25)	12 (25)
Diagnosis of early onset sepsis	13 (36)	1 (8)	14 (29)
Clinical and Biological Signs: n (%) or Median (IQR)			
Cardio-respiratory events	31 (86)	12 (100)	43 (89)
Abnormal temperature	26 (72)	4 (33) *	30 (62)
Abnormal peripheral hemodynamic	20 (55)	2 (17) *	22 (45)
Amines or volume expansion	11 (31)	3 (25)	14 (29)
Heart rate >180/min	13 (36)	6 (50)	19 (40)
Respiratory rate >80/min	23 (64)	4 (33) *	27 (56)
Fi02 max,%	29,5 (21–48.5)	32 (21–55)	30 (21–48)
PEP max, cm H20	6 (5–6)	5 (5–6)	6 (5–6)
SCORE Fi02-P increased	12 (33)	4 (33)	29 (33)
Hyperglycemia	19 (53)	4 (33)	23 (48)
Digestive signs	14 (39)	3 (25)	17 (35)
No. of associated signs	4.5 (3–6.5)	4 (2.5–4)	4 (3–5.2)
Feeding stop	19 (53)	4 (33)	23 (48)
CRPmax (mg/L)	24 (11–46)	2 (0–6)	17 (5–40)
CRPmax >10 mg/L	29 (80)	1 (8) *	30 (63)
Start antibiotic therapy, days	8.1 (5.6–10.4)	6.2 (4.8–6.2)	7.5 (5.4–10.5)
BPD or Death, *n* (%)			
BPD or death	28 (78%)	3 (25) ***	31 (65)
BPD	19 (53)	3 (25)	22 (61)
Death	9 (25)	0 (0)	9 (25)

* *p* < 0.05, ** *p* < 0.01, *** *p* < 0.001: Mann–Whitney U test between “certain or probable” or “doubtful” LOS groups. FiO2: inspired fraction of oxygen; PEP: positive end expiratory pressure; CRP: C reactive protein; BPD: bronchopulmonary dysplasia.

**Table 2 ijerph-18-00882-t002:** Concordance study of the international US Centers for Disease Control and Prevention (CDC)/Canadian Neonatal Network (CNN)/ German National Reference Centre for Nosocomial Infection Surveillance in Neonatal Intensive Care Units (NEO-KISS) classification of the adjudication committee in 2 groups: Cohen’s kappa and rate of agreement matrix.

Kappa Cohen (Rate of Agreement)	NEO-KISS	CNN	Adjudication Committee
CDC	0.12 (42)	0.03 (31)	0.22 (52)
NEO-KISS		0.41 (90)	0.42 (81)
CNN			0.23 (79)

**Table 3 ijerph-18-00882-t003:** Inter-observer concordance study between committee experts and National Institute for Child Health and Human Development (NICHD). NICHD classify the late-onset bacterial sepsis (LOS) in three groups as the expert did.

Kappa Cohen (Rate of Agreement)	Expert 2	Expert 3	Expert 4	Expert 5	NICHD
Expert 1	0.53 (71)	0.45 (65)	0.58 (73)	0.51 (69)	0.4 (63)
Expert 2		0.35 (58)	0.32 (56)	0.45 (67)	0.3 (56)
Expert 3			0.33 (56)	0.33 (56)	0.19 (48)
Expert 4				0.3 (54)	0.29 (54)
Expert 5					0.28 (56)

**Table 4 ijerph-18-00882-t004:** Heart rate variability (HRV) parameters: “certain or probable” vs. “doubtful “LOS between T0 − 6 h and T0 + 6 h. The results are therefore expressed as Z scores (i.e., ((observed value—mean of the reference period)/standard deviation from the reference period)).

HRV Features (Z Score)	Doubtful LOS Median (IQR) (*n* = 10)	LOS Certain or Probable Median (IQR) (*n* = 30)	CDC (*n* = 9)	Neo-Kiss (*n* = 38)	CNN (*n* = 34)
Mean RR	0.18 (−2.54, 1.46)	−0.5 (−1.04, 0.87)	NS	NS	NS
rMSSD	−0.28 (−0.47, 0.04)	−0.38 (−0.75, 0.08) *		*	*
Kurtosis	−0.12 (−0.32, 0.3) §	−0.35 (−0.51, −0.11) **	*	**	**
HF	−0.31 (−0.39, −0.12)	−0.42 (−0.75, −0.19) **		**	**
HFnu	−0.3 (−0.56, 0.8) §	−0.54 (−0.8, −0.41) **	*	**	**
SD1	−0.28 (−0.47, 0.04)	−0.4 (−0.75, 0.08) *		*	*
SampEn	−0.58 (−0.94, −0.51) *	−0.82 (−1.08, −0.29) *		*	*
ApEn	−0.56 (−0.89, −0.16)	−0.69 (−1.21, −0.18) *		**	**
Alpha 1	0.62 (−0.09, 0.77)	0.4 (0.1, 0.85) **		**	**
Tr VG	−0.04 (−0.18, 0.6)	−0.52 (−1.54, 0.19) *		*	*

* *p* < 0.05 vs. control period ** *p* < 0.05 vs. control period after Bonferroni correction; § *p* < 0.05 vs. certain or probable. Mean RR: mean duration of cardiac cycle length; rMSSD: root-mean-square of successive RR intervals; HF: high-frequency power with limits 0.2–2 Hz; HFnu: HF in normalized units; SD1: short-term heart rate variability derived from the Poincaré plot analysis; SampEn: sample entropy; ApEn: approximate entropy; Alpha1: short-term fractal coefficient (regularity); Tr-VG: transitivity in vertical visibility studies. CDC, NEO-KISS, CNN: the significance of changes in HRV with CDC, NEO-KISS, and CNN definitions; NS: non-significant.

## Data Availability

The data presented in this study are available on request from the corresponding author.

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
