# Peer review of "Diagnosis of Neonatal Late-Onset Infection in Very Preterm Infant: Inter-Observer Agreement and International Classifications"

_ijerph, 2021, doi:10.3390/ijerph18030882_

Round 1

Reviewer 1 Report

Summary:

The group of Bury et al aims to describe the current inconsistency in late onset sepsis (LOS) definition among preterm infants. They perform a retrospective multicenter study, consulting five experts of the field to classify LOS cases in preterm infants <31 weeks gestational age as either “certain or very probable” “probable” or “doubtful” based on different international classifications. The primary identification of LOS cases from the prospective cohort study Digi-NewB were more than five days of antibiotic treatment (or death before completion of the course) with start of the treatment at least 72 hours after birth. They additionally assessed association of heart rate variability (HRV), rate of bronchopulmonary dysplasia (BPD) and death with different LOS classifications.

This topic is of major importance as different definitions makes comparison of clinical work and research very difficult and hence limits progress.

Some improvements / changes are suggested before possible acceptance for publication:

1: Line 132: “Bradycardia was defined as heart rate <100 beats/ min for >5 seconds, or <80 beats/ min for >10 seconds” => If this definition is true, this would imply to different levels of bradycardia (first one less severe, shorter and second one more severe and longer), did you adjust for that fact?

2: Concerning he FiO2-P score: did the score include changes in level of respiratory support as e.g. start of CPAP or change from non-invasive to invasive ventilation? Please specify.

3: How much time apart from LOS suspicion was allowed for NEC diagnosis to be included in the outcome table (see line 149 and following). Please specify.

4: Consider summarizing different considered clinical factors and finally included ones in the analysis in a table. At the moment it is hard to follow e.g. line 156 describes “unusual episode of hyperglycaemia” which is not further specified until in line 159 it is mentioned again and specified there.

5: To give the reader an overview of different time-windows assessed before and after T0, an overview graph might be helpful (e.g. period around infection was defined as five days before and five days after start of antibiotic treatment (T0), T0-12 until T+60h for biological variables, HRV variables T0 plus/ minus 6h, reference values calculated over period of 12h outside of infectious episode (at least three days before or four days after T0)).

Justify why the windows were different in size for different parameters/ analysis if the period around T0 was initially set to +/- 5 days.

6: Line 169 and following: was a standardized room-air test performed to ascertain diagnosis of BPD at 36 weeks gestation? Or was information drawn retrospectively from clinical journals? If so, was it someone of the study team that accessed the journal and if so, was this person blinded to the sepsis-group the patient was assigned to? Please clarify.

7: Figure 1: please provide more detailed information about the Figure: what does the blue line and dotted orange line represent? Provide description of axis with corresponding units.

8: Which level of significance alpha was set after the Bonferroni correction? Please specify as several degrees can be implemented. Additionally, note that it is called “Bonferroni” with 1 n only.

9: Did you assess how many (%) of infants did have two or three of the collected clinical signs in control periods (periods when not suspected of sepsis) as compared to 98 and 55% respectively during sepsis episodes? This would be of interest to show the possible differences upon sepsis development, as per se all infants included in the study are patients in need of intensive care treatment and are known to have signs like bradycardia, apnea, dependency of respiratory support, … anyway.

10: The discussed bias in the study by Jung et al, showing increased risk of BPD with higher number of LOS episodes could also be present in the current study, please elaborate on this thought in the discussion.

Minor comments:

  • Line 100: use abbreviation VPT introduced before
  • Line 114 repeats description of CLABSI, CLASS characterization of catheter-related infections already described in line 106 and following.
  • Line 138 and following: The description of “FiO2-P score” is repeated in two consecutive sentences. Delete one of them.
  • Line 152: please specify EDIN and Neocomfort scale with a suitable literature reference.
  • Line 158 and following: CRP was abbreviated before and can be put in short form here only. Procalcitonin was written with a small letter at the start before and here is written with a capital letter, please be consistent.
  • Line 185: post-hoc classified instead of pot-hoc classified
  • Line 196: introduce RR abbreviation when first used and not in sentence on line 197.
  • Line 201 and following: “ … which characterize the acceleration and deceleration capacity of the heart rate,
  • Line Line 205 until 210: suggested to write in 2-3 sentences to make it easier for the reader.
  • Line 2018: “All the measurements made were standardized and expressed…”
  • Decide if Mann and Whitney´s U or u test (no both used)
  • Delete manuscript guidelines in Results section (line 246-248) and for tables (line 317 and following)
  • Delete “Clinical signs are presented in Table 1.”, line 261 as already written in line 255.
  • Line 263: Specify if cardiorespiratory events were more prominent 12 hours after start of antibiotics or until 12 hours after the termination of antibiotic treatment.
  • Line 265: FiO2
  • Line 267: enterocolitis => NEC (as introduced before and more specific)
  • Line 285: abbreviation SCNs not introduced before
  • Line 290: “: 13 patients were considered having LOS according to ….”
  • Standardize Neokiss vs Néokiss within manuscript and tables
  • Line 306: vs instead of Vs
  • Line 311: “CRP was also higher in the infected group…” consider not to call them infected group, as the whole manuscript is about difficulties in defining who is infected.
  • Within table 1: be consistent whether to write capital or small letters at beginning (e.g. Abnormal Temperature vs Abnormal peripheral haemodynamic), define PEP description (mean? max?)
  • Lines 322 and following: Text seems to belong to table 1 and not table 2. Within that text: CRP: C-reactive protein
  • Line 384: delete (14, 21, 32-37) or include in references if intended so
  • Line 395: “In the neonatal period and with current methods…”
  • Line 396: “In our study we only identified six infants with …”
  • Line 402: “Coag- neg Staph. was identified on blood cultures ….”
  • Line 413: “because of their low diagnostic value at the time of suspicion. For example, CRP levels may not increase, or…”
  • Line 419: abbreviation MDT was not introduced before
  • Line 424: “nSOFA score”
  • Line 429: please specify if significant differences also existed for both groups within each group when comparing sepsis episodes with episodes where no sepsis was suspected or if there were significant differences between the groups when compared the two in periods where no sepsis was suspected.
  • Line 432: rephrase: “… as a physiological marker of LOS with a loss of complexity in HRV at the early stage…”
  • Line 440: abbreviations SA and OR were not introduced before.
  • Line 450: rephrase sentence “… the result of the current study, we think with others that ….”
  • Reference list: many references are not completed and have missing doi, e.g. refs 2, 8, 15, 24

Author Response

RE:         Bury G et al: Diagnosis of neonatal late-onset infection in very preterm infant: inter-observer agreement and international classifications. IJERPH-1033799

Dear Dr. Editor,

We thank the reviewers and yourself for having reviewed our manuscript and for the constructive comments that help to clarify our message. Attached please find the detailed responses to the reviewers and our revised version of the manuscript (a version with the changes marked and a “clean copy”, as requested). The answers to the three reviewers have been uploaded together in the attached word document.

We hope the manuscript is now acceptable for publication in the Journal.  Thank you for your consideration.

Sincerely,

Pr Patrick Pladys, chef de pôle-Chef de service,  CHU Rennes

Pôle Femme Enfant, Service de pédiatrie

 INSERM U1099, CIC1414 et Réseau pédiatrique de recherche du Grand Ouest HUGOPEREN

Mobile : ( 33) 6 34 19 11 30

https://www.digi-newb.eu/

http://www.ltsi.univ-rennes1.fr/

http://www.hugoperen.org/

Reviewer 2 Report

Diagnosis of neonatal late-onset infection in very preterm infant: inter-observer agreement and  international classifications

The definition of late onset bacterial sepsis in very preterm infants is not well unified. This study point out important limits and heterogeneities in the current definition of late onset bacterial sepsis and confirms the low diagnostic values of clinical, biological, and microbiological criteria for early diagnosis of late onset bacterial sepsis. This work is a good contribution to try to obtain unified detection criteria and thereby improve care and treatment.

Suggestions to correct

Lines 246-8. Remove all

Lines 317-8: Remove all

Line 330. In Table 4 (final point)

Line 348. Remove (14,21,32-37)

Author Response

RE:         Bury G et al: Diagnosis of neonatal late-onset infection in very preterm infant: inter-observer agreement and international classifications. IJERPH-1033799

Dear Dr. Editor,

We thank the reviewers and yourself for having reviewed our manuscript and for the constructive comments that help to clarify our message. Attached please find the detailed responses to the reviewers and our revised version of the manuscript (a version with the changes marked and a “clean copy”, as requested). The answers to the three reviewers have been uploaded together in the attached word document.

We hope the manuscript is now acceptable for publication in the Journal.  Thank you for your consideration.

Sincerely,

Pr Patrick Pladys,

Rewiewer 2 : Suggestions to correct

Lines 246-8. Remove all

Lines 317-8: Remove all

Line 330. In Table 4 (final point)

Line 348. Remove (14,21,32-37)

Reviewer 3 Report

  1. This paper points out the inconsistency of the different criteria in LOS diagnosis. And found several sensitive clinical signs that may help characterize LOS. Based on these signs (i.e. clinical signs, HRV), is it possible to build an objective classification model to improve the diagnosis accuracy?
  2. For Table 1, cardio-respiratory events was 43, but in the manuscript it was 34.
  3. For Table 1, PCT information is missing?
  4. What are the potential reasons for the inconsistency of different international classifications? Some short description should be placed in supporting material or main text to provide how these test works, and what are the major parameters.  On the other hand,  NeoKISS and CNN share high agreement with each other and experts group, do they cover some specific and meaningful parameters (compared with CDC) in their tests? Should we conclude NeoKISS and CNN should be preferred to use?
  5. To evaluate the robustness of the importance of HRV (and/or other clinical signs), how would the comparison of HRV between groups changes if using different international classification method. Would the findings be consistent with the reported results based on experts’ classification?

Author Response

RE:         Bury G et al: Diagnosis of neonatal late-onset infection in very preterm infant: inter-observer agreement and international classifications. IJERPH-1033799

Dear Dr. Editor,

We thank the reviewers and yourself for having reviewed our manuscript and for the constructive comments that help to clarify our message. Attached please find the detailed responses to the reviewers and our revised version of the manuscript (a version with the changes marked and a “clean copy”, as requested). The answers to the three reviewers have been uploaded together in the attached word document.

We hope the manuscript is now acceptable for publication in the Journal.  Thank you for your consideration.

Sincerely,

Pr Patrick Pladys, c

Rewiewer 3 :

This paper points out the inconsistency of the different criteria in LOS diagnosis. And found several sensitive clinical signs that may help characterize LOS. Based on these signs (i.e. clinical signs, HRV), is it possible to build an objective classification model to improve the diagnosis accuracy?

For Table 1, cardio-respiratory events was 43, but in the manuscript it was 34

We have clarified this in the manuscript: Cardiorespiratory events included apnea and/or bradycardia and/or desaturation. They were noted in 43 infants. They occurred or increased within 24 hours before and 12 hours after start of antibiotics in 34 of cases (69%). “

For Table 1, PCT information is missing?

We didn’t presented the results of PCT in Table 1 because PCT was available for only 27 patients. The results are presented in the text of the manuscript.

What are the potential reasons for the inconsistency of different international classifications? Some short description should be placed in supporting material or main text to provide how these test works, and what are the major parameters.  On the other hand,  NeoKISS and CNN share high agreement with each other and experts group, do they cover some specific and meaningful parameters (compared with CDC) in their tests? Should we conclude NeoKISS and CNN should be preferred to use?

We have modified the discussion as follows to answer to the reviewer comment and question:

“The potential reasons for the observed inconsistency of the different international classifications in our study are numerous. Variations in the international classifications imply clinical criteria used in the definition of sepsis, time at which sepsis occur, interpretation of contaminant or polymicrobial cultures, pathogen isolated from other site than blood or CSF, use of duration of therapy as a criterion for the diagnosis, cut-off values for sepsis biomarkers, and subclassification of sepsis with regard to NEC, pneumonitis or meningitis. For example the clinical  presentation of  what  has  been  termed  NEC  can  be  highly  variable,  with  signs  and  symptoms  that are often indistinguishable from sepsis. In our study We didn’t introduce specific rules to distinguish between LOS and NEC. One of the consequences of this is that comparison between published studies is often difficult or unreliable. In our study the best concordance between the adjudication committee and international classifications was obtained with the NEO-KISS classification with a moderate concordance and an agreement rate of 81%. In addition, we observed a similar impact on heart rate variability and the rate of "bronchopulmonary dysplasia or death" with both the adjudication committee and NEO-KISS classifications. Therefore, it appears that NEO-KISS is likely to be an appropriate classification to use for the neonatal units involved in the study.  ”

To evaluate the robustness of the importance of HRV (and/or other clinical signs), how would the comparison of HRV between groups changes if using different international classification method. Would the findings be consistent with the reported results based on experts’ classification?

Following the suggestion of the reviewer we have studied the changes in HRV and in the rate of bronchopulmonary Dysplasia or death. These results have been integrated in the results

In the text of the results we have added the following sentence: “The rate of "bronchopulmonary dysplasia or death" was also significantly higher with the NEO-KISS classification [n=29 (71%) vs 2 (29%), p<0.05, Odds Ratio = 6 with CI95%: 1.02-36] but not with the other classifications studied.”

The significant changes in HRV are also presented in Table 4:

  • CDC (n=9): Kurtosis (p=0.02); HFnu (p=0.02)
  • Neo-Kiss (n=38): rmssd (p =0.02); Kurtosis (p=0.0002); HF (p =0.002); HFnu (p=0.0008); SD1 (p=0.02); SampEn (0.004); ApEn (0.002); alpha1 (0.0002); Tr VG (p=0.04)
  • CNN (n=34): rmssd (p =0.02); Kurtosis (p=0.0006); HF (p =0.001); HFnu (p=10-6); SD1 (p=0.01); SampEn (O.007); ApEn (0.002), alpha1 (0.001); Tr VG (p=0.01); MD-HVG (p=0.02)